# Abiotic and Biotic Influences on the Movement of Reintroduced Chinese Giant Salamanders (*Andrias davidianus*) in Two Montane Rivers

**DOI:** 10.3390/ani11061480

**Published:** 2021-05-21

**Authors:** Qijun Wang, Lu Zhang, Hu Zhao, Qing Zhao, Jie Deng, Fei Kong, Wei Jiang, Hongxing Zhang, Hong Liu, Andrew Kouba

**Affiliations:** 1College of Fisheries, Key Lab of Freshwater Animal Breeding, Ministry of Agriculture and Rural Affair/Key Lab of Agricultural Animal Genetics, Breeding and Reproduction of Ministry of Education, Huazhong Agricultural University, Wuhan 430070, China; wqjab@126.com; 2Shaanxi Institute of Zoology, 88 Xingqing Road, Xi’an 710032, China; zhaohu2007@126.com (H.Z.); Dengjie0311@ms.xab.ac.cn (J.D.); k.coffee@163.com (F.K.); jiangwei197981@163.com (W.J.); zhs@ms.xab.ac.cn (H.Z.); 3School of Life Sciences, Sun Yat-sen University, Guangzhou 510275, China; zhanglu38@mail.sysu.edu.cn; 4Department of Wildlife, Fisheries and Aquaculture, Mississippi State University, Starkville, MS 39762, USA; 5School of Natural Resources, University of Missouri, Columbia, MO 65211, USA; whitelangur@gmail.com

**Keywords:** body condition, demographics, moon phase, movement, precipitation, reintroduction, temperature, tracking

## Abstract

**Simple Summary:**

The movement of critically endangered Chinese giant salamander (*Andrias davidianus*) remains poorly understood due to the rareness of wild individuals. We examined the impacts of individual traits and environmental conditions on daily movement patterns of reintroduced giant salamanders. We found that movement tendency of the older salamander cohort was positively affected by the moon phase, but the moon phase did not impact the younger cohort of animals. For daily distance moved, we found temperature had a strong positive effect on both cohorts, whereas precipitation had moderate but opposite effects on the two cohorts. Body mass and body condition did not have any significant impacts on either age classes’ movement tendency or distance. This study helps to fill in knowledge gaps on the movement ecology of reintroduced Chinese giant salamanders, which will benefit the design of future reintroduction efforts for this critically endangered species.

**Abstract:**

Understanding animal movement is a key question in ecology and biodiversity conservation, which is particularly important for the success of reintroduction projects. The movement of critically endangered Chinese giant salamander (*Andrias davidianus*) remains poorly understood due to the rareness of wild individuals of this species. We lack movement details about the full annual cycle after reintroduction, especially the abiotic and biotic influences that affect its movement. We utilized pilot reintroduction projects as opportunities to fill in some knowledge gaps on their movement ecology. We released 31 juvenile captive-reared Chinese giant salamanders of two age groups in two rivers in the Qinling Mountains of central China and monitored their daily movements for 16 months using surgically implanted radio transmitters. We examined the impacts of individual traits (body mass, body condition) and environmental conditions (temperature, precipitation, and moon phase) on their daily movement patterns. Data were analyzed using a mixed-effects logistic regression model to understand the drivers of their movement tendency (i.e., whether they move or not) and a linear mixed-effects model was used to understand the drivers of their movement distance. We found that movement tendency of the older salamander cohort was positively affected by the moon phase, increasing near the Full Moon, whereas the younger cohort of animals were not impacted by the moon phase. For daily distance moved, we found temperature had a strong positive effect on both cohorts, whereas precipitation had moderate but opposite effects on the two cohorts Body mass and body condition did not have any significant impacts on either age classes’ movement tendency or distance. This study provides insight into the abiotic factors that impact the temporal and spatial movement ecology of reintroduced giant salamander, which will in turn help with designing best practices for future releases and conservation of this iconic montane aquatic predator.

## 1. Introduction

Understanding animal movement is a key question in ecology and biodiversity conservation [1,2,3]. In particular, the success of reintroduction often relies highly on the knowledge of the target species’ movement ecology [4]. Reintroduction is increasingly used as a conservation tool to restore species to their historical range [5,6]. Reduced movement and dispersal of reintroduced or translocated animals may help with the initial population establishment in the wild and enhance their sustainability over time [5,7]. In fact, one of the most reported causes of translocation/reintroduction failure was that introduced individuals move out of release sites [8]. Hence, understanding the movement ecology of the target species is an important consideration for the success of reintroduction projects [4].

The movement of animals is often determined by their internal state, motion capacity, navigation ability, and external factors [1]. However, the effectiveness of individual factors seems to be species-specific. Internal factors such as age, sex, and body mass are found to affect animals’ movement patterns [9,10,11], although other studies reveal no significant relationship [12]. Similarly, external factors including temperature, precipitation, and the moon phase are also found to affect the movement pattern of some species but not others [13,14,15], even for species that are relatively close-related. For example, leopard tortoises (*Stigmochelys pardalis*) move more at higher temperatures, whereas Eastern box turtles (*Terrapene carolinacarolina*) were found to be a thermal generalist [16]. Empirical studies are needed to reveal species-specific relationships between internal and external factors and the movement pattern of the species.

The Chinese giant salamander (*Andrias davidianus*) is fully aquatic, can reach up to 1.8 m in length, and is recognized as the largest living amphibian species in the world [17,18]. Once widely distributed in tributaries of the Yellow, Yangtze, and Pearl Rivers in central/southern China [19], the population of the Chinese giant salamander has declined over 80% since the 1950s due to habitat destruction, water pollution, and human consumption [18]. These continuing threats resulted in the species being listed as “Critically Endangered” and a Class Ⅱ protected species in China [18,20]. As an apex or mesopredator in first to third-order streams, it consumes a wide array of prey items, including crabs, fish, snakes, other amphibians, aquatic insects, water birds, and small mammals [21,22]. Thus, declining of extant populations of giant salamander could have profound impacts on the trophic ecology within these aquatic systems.

While large-scale commercial breeding farms and poaching have dramatically impacted wild giant salamander populations [23], the reproductive success of these commercial units presents them to be source populations for recovery efforts. Moreover, the large number of offspring produced annually by these farms suggests reintroduction efforts may be more feasible and sustainable than for other declining amphibian species that do not have a broodstock population [24,25].

Unfortunately, very little is known about wild Chinese giant salamander, with significant knowledge gaps in its movement ecology. The rapid and large-scale decline and rareness of this species in the wild make it difficult to study these animals’ natural behaviors. So far, no published works have revealed any information on the movement ecology of wild Chinese giant salamander, unlike the closely related hellbender (*Cryptobranchus alleganiensis*) [26,27,28] or Japanese giant salamander (*A**. japonicas*) [29]. Although reintroduced individuals may behave differently from their wild conspecifics, studies of reintroduced or translocated giant salamanders can still fill in important knowledge gaps and provide a reference for future studies (e.g., [30,31]).

We released 31 juvenile captive-reared giant salamanders at two sites in the Qinling Mountains of central China and monitored them daily for 16 months through radio telemetry. Previously we have reported post-release survival, habitat selection, and some basic information on the movement ecology of these salamanders, including sedentariness, daily displacement, home range, and dispersal [24,31,32]. However, the environmental and individual drivers that influence their movement tendency and distance remain unclear. A better understanding of these abiotic and biotic factors can assist in developing better management practices for future reintroductions.

In this study, we examined the impacts of two individual traits (body mass and body condition) and three environmental variables (temperature, precipitation, and moon phase) on the daily movement of giant salamanders. We hypothesized that giant salamanders would move more frequently and for longer distances as temperature, precipitation, and moon phase increased, similar to how these variables affect the movement of other amphibians [33,34,35,36]. We also hypothesized that giant salamanders with larger body mass and better body condition would be able to move more frequently and over longer distances [37]. We aim to fill in knowledge gaps on the movement ecology of reintroduced Chinese giant salamanders, which will benefit the design of future reintroduction efforts for this critically endangered species.

## 2. Materials and Methods

### 2.1. Study Area and Animals

This reintroduction study was conducted in two rivers in the Qinling Mountains in central China (Figure 1). 

The Heihe River (33°53′ N, 108°00′ E, Zhouzhi County, Xi’an, China) is on the north slope of the Qinling Mountains and belongs to the Yellow River watershed, with deciduous broad-leaf forest as the main vegetation type along the river (Figure 2A). The Donghe River (33°21′ N, 108°16′ E, Ningshan County, Xi’an, China) is on the south slope of the Qinling Mountains and belongs to the Yangtze River watershed, with a mixture of evergreen and deciduous broad-leaf forest as the main vegetation type along the river (Figure 2B). Although rare, wild giant salamanders are still occasionally found in these two rivers, indicating they continue to provide a relatively good habitat for giant salamanders.

The study involved 31 captive-reared juvenile giant salamanders. All animals were surgically implanted with radio transmitters (F1035, Advanced Telemetry Systems, Inc., Isanti, MN, USA) before release, as previously described [38]. The younger cohort of animals (*n* = 15) were collected as larvae from the Heihe River, head-started in a commercial breeding farm, and released at 3 years of age (0.5 ± 0.2 kg; 44.00 ± 3.24 cm total length) at the Heihe River. Fourteen of them were released on the 28 April to 2 May 2013 plus one additional salamander was released on 5 November 2013. The older cohort of animals’ parents were collected as juveniles in the Donghe River, bred in a commercial breeding farm, and a cohort of offspring (*n* = 16) were released at 5 years of age (1.6 ± 0.4 kg; 63.97 ± 4.86 cm total length) back into the Donghe River on 12 July 2013. The rearing conditions were comparable between younger and older cohorts at the two farms. They were reared in concrete tanks with many conspecifics and were provided with adequate food (details can be found in [39]). Captive-reared giant salamanders usually reach adulthood at 8 years old [40]. Our study animals were juveniles and several years away from sexual maturity, even for the older cohort. These salamanders were released in a ~50 m long river section at both sites and each animal was released beside a rock large enough to provide shelter.

### 2.2. Movement Monitoring and Abiotic Data Collection

All giant salamanders were monitored daily, post-release, using a radio receiver with a 3-element Yagi antenna (R410, Advanced Telemetry Systems, Inc., Isanti, MN, USA) until the battery of the radio transmitters died. Salamander location coordinates were recorded by handheld GPS units (60CSx, Garmin, Ltd., New Taipei City, Taiwan). In addition to the authors, trained local field assistants helped with tracking and monitoring of released salamanders throughout the year. As boulders selected by the animals for shelter were usually too large to turn over physically, we determined the presence of salamanders using an underwater inspection camera (M12, Milwaukee Electric Tool, Brookfield, WI, USA). 

We placed data logger probes (HOBO Water Temp Pro v.2; Onset Computer Corporation, Bourne, MA, USA) at both rivers to collect hourly water temperature information. We also downloaded daily precipitation data from the China National Meteorological Information Center (http://data.cma.cn/, accessed on 6 April 2016), with a 0.5° latitude/longitude accuracy to location. Moon phase data, ranging from 0 (New Moon) to 1 (Full Moon), were obtained from the United States Naval Observatory (http://aa.usno.navy.mil/data/docs/MoonFraction.php, accessed on 29 March 2016).

### 2.3. Statistical Analysis

To support comparison among giant salamanders, daily movement was calculated in ArcGIS (Version 10, [41]) as the straight-line distance (m) between locations collected in sequential days for each animal. Movements with ≤3 m distance were considered as stationary, since the accuracy of the GPS unit that we used was 3–5 m.

We conducted two sets of regression models to analyze the impacts of body mass, body condition, temperature, precipitation, and moon phase on salamanders’ daily movement. A mixed-effects logistic regression model was used to determine the drivers of movement tendency, with daily displacements >3 m as ‘moved’ and daily displacements ≤3 m as ‘did not move’. Moreover, a linear mixed-effects model was used to determine the drivers of movement distance (for daily displacement >3 m), with individual ID as a random effect. Body condition was calculated as the residual to the regression line of the cubed root of mass and total length constructed using all 31 salamanders before release [42]. Daily water temperature was obtained by averaging hourly water temperature at both rivers. Considering a possible lag between precipitation and the rise of water volume and velocity in the rivers, we also tested the impact of precipitation from the previous day on salamanders’ displacement. 

We first tested the multicollinearity among the six independent variables and found the Pearson’s correlation coefficients ranged from 0.000 to 0.481, indicating that no independent variable needed to be removed. We then conducted model selection based on AIC to select the best model explaining daily movements. The six independent variables were added to the null model, respectively, to compare their AICs with that of the null model. All combinations of the variables that improved model support (with reduced AIC value compared to the null model) were also tested. The model with the minimum AIC value was considered to be the best model fit and models with ≤2 ∆AIC were considered as having equivalent support [43]. All statistical analyses were conducted in R (version 3.3.3 [44]), using the packages lme4 [45] and usdm [46].

## 3. Results

A total of 5491 records of daily movements were collected during our monitoring period, among which 4252 records were from the older cohort of salamanders reintroduced at the Donghe River, and 1239 records were from the younger cohort reintroduced at the Heihe River. The mean daily distance moved was 9.3 ± 0.3 m (*n* = 3185, 4–880 m) at the Donghe River, and 15.4 ± 0.7 m (*n* = 979, 4–298 m) at the Heihe River [31]. At the Donghe River, water temperature ranged from 0 to 23.9 °C during the study period, whereas water temperature at the Heihe River ranged from 0 to 21.0 ℃.

For the older cohort, the best-supported model explaining salamander movement tendency only contained one variable—moon phase, with a coefficient of 0.271 ± 0.103 (Table 1). Salamanders tend to move more frequently during the period around the Full Moon but less frequently around the New Moon. For the younger group, none of the variables had a significant impact on salamander movement tendency (Table 1). When salamanders did move, the distance moved was positively affected by temperature for both cohorts (Table 2). Salamanders of both cohorts moved longer distances at higher temperatures, with a coefficient ± SE of temperature at 0.115 ± 0.020 for the older cohort and 0.578 ± 0.183 for the younger cohort. For the older cohort, a model with both temperature and precipitation on the same day (coefficient ± SE = 0.066 ± 0.033) as predictors of movement distance had equivalent support as the model with the temperature only (∆AIC = 1). Similarly, for the younger cohort, a model with both temperature and precipitation in the previous day (coefficient ± SE = −0.103 ± 0.069) had equivalent support as the model with the temperature only (∆AIC = 0). Overall, the results indicate a strong effect of temperature and a moderate effect of precipitation on the movement of Chinese giant salamanders. We did not find evidence for body mass or body condition influencing movement tendency or distance for either cohort of released salamanders.

## 4. Discussion

We used data from an intensive one-year telemetry study on captive-reared Chinese giant salamanders reintroduced to the wild to examine their post-release movement pattern in relation to individual and environmental drivers. We found that the older cohort of salamanders moved more frequently nearer a Full Moon, whereas the moon phase did not seem to impact the younger cohort’s movement. When salamanders did move, they tended to move longer distances under higher water temperature for both age groups. Precipitation had a moderate effect on salamanders’ daily displacement. Interestingly, precipitation had a positive effect on the older group, while having a negative effect on the younger group. Biotic drivers examined in our study, i.e., mass and body condition, were not found to have a significant impact on the movement pattern of reintroduced giant salamanders.

The effect of the moon phase on animal movements has been studied on a wide range of taxa [47]. Behavioral responses to moonlight are quite species-specific, even within the same taxon [48] and are often related to predator–prey interactions [49]. Predators might increase their activity levels when the moon is present, taking advantage of the increased visibility of potential prey [50,51]. Conversely, prey may be more active when they are less visible to predators [52,53]. The adult Chinese giant salamanders are predators in the montane stream systems, although they could be predated upon by some small carnivores, such as Eurasian otters (*Lutra*
*lutra*). Thus, giant salamanders are more likely to be a meso predator and their movement tendency is a trade-off between predator avoidance and prey acquisition when other top predators are present, and their movement related to moon phase may be modified by age-related vulnerability to top predators. We found the older cohort in our study (at the Donghe River) were more likely to move as the moon phase increased, which is probably due to the fact that they are older and less vulnerable to predators. By contrast, the younger cohort (at the Heihe River) did not move more frequently as moon phase increased, which may represent the fact that they are younger and are thus balancing prey acquisition against the risk of encountering predators [54]. Studies that examine the distribution of potential predators (e.g., otters, [55]) and the changes in juvenile salamander movement patterns over time are needed to further test these hypotheses. Furthermore, the fact that the two cohorts were hatched differently (i.e., wild hatched vs. captive hatched), reared in different farms (although with similar rearing conditions), and were reintroduced at two sites which differed in many aspects such as river width, flow rate, boulder density, and prey abundance may confound the effect of age. Future studies need to consider using salamanders with similar rearing background and reintroducing them at the same site to clarify the effect of moon phase on different age groups.

Water temperature was found to positively influence salamander movement distance for both age groups, which is consistent with many studies on ectotherms, for example, Viviparous lizard (*Lacerta vivipara*) [56] and barbel (*Barbus*
*barbus*) [57]. Temperature has long been found to affect animal metabolism, such that higher temperatures lead to increased activity and velocity, which is especially true for ectoderms [58,59]. The thermal preference of Chinese giant salamanders in the wild has not been reported yet, although some studies in captivity reveal that giant salamanders are normally active when water temperatures are near 24–25 °C and reduce activities when temperatures become elevated [21,60,61]. Water temperature at both rivers was below 24 °C all year round. Hence, these giant salamanders are likely to move longer distances at both sites as temperature increased.

We initially proposed giant salamander movement may adhere to the precipitation hypothesis, primarily based on studies for pond-breeding amphibians, where large movement events would be associated with precipitation [33,62]. However, as a fully aquatic amphibian, giant salamanders may respond differently to precipitation compared to pond-breeding amphibians. In riverine systems, precipitation would cause fluctuation of river flow and depth. The relationship between flow and fish movements has been carefully studied, and it was found that river flow usually had a positive effect on both movement frequency and distance moved [63]. However, for stream-dwelling salamanders, most studies focus on evaluating the impact of river flow on their dispersal (e.g., [64,65]), or do not relate salamander movement to water flow/precipitation (e.g., [34]). The relationship between river flow/precipitation and salamander movement has not been well studied. We propose that salamanders may be more resistant to river flow than fish, such that precipitation had a relatively weak positive effect on the older group of giant salamanders. The negative impact of precipitation on the younger group of salamanders requires further studies. However, similar to pond-breeding amphibians [35], extreme variation in precipitation related to drought or flood events may have a larger impact on giant salamanders, such that we recorded several long-distanced downstream movements during extreme precipitation events that induced flooding [31].

Animals with larger body mass and in better condition are assumed to have more energy reserves [66], thus be able to move more frequently and for longer distances [31,67]. However, our results indicated no relationship between body mass or condition and daily salamander movement, which might suggest that because salamanders were captive-reared together, intra-cohort differences were not large enough to have a significant impact on their movement capacity. We did not consider inter-cohort differences on their daily movement, since the two release sites differed in many aspects (e.g., river width, flow rate, boulder density, and prey abundance), which may affect the movement behaviors of salamanders. Future studies on reintroduction of giant salamanders should evaluate different age groups and body conditions at the same site, which may help to clarify the relationship between body mass or condition on daily movement.

## 5. Conclusions

In conclusion, three important environmental drivers of movement were identified for reintroduced Chinese giant salamanders. We found the moon phase had a positive effect on movement tendency of the older cohort of giant salamanders, yet had no effect on the younger cohort. For daily distance moved, we found temperature had a strong positive effect on both cohorts, whereas precipitation had moderate but opposite effects on the two cohorts, which requires further studies to clarify. In a rejection of our hypotheses, body mass and body condition were not found to have a significant impact on salamander daily movement. Our study provides insight into the abiotic factors that affect the daily movement patterns of reintroduced giant salamander, which will, in turn, help with designing better practices for future releases and conservation of this iconic montane aquatic predator.

## Figures and Tables

**Figure 1 animals-11-01480-f001:**
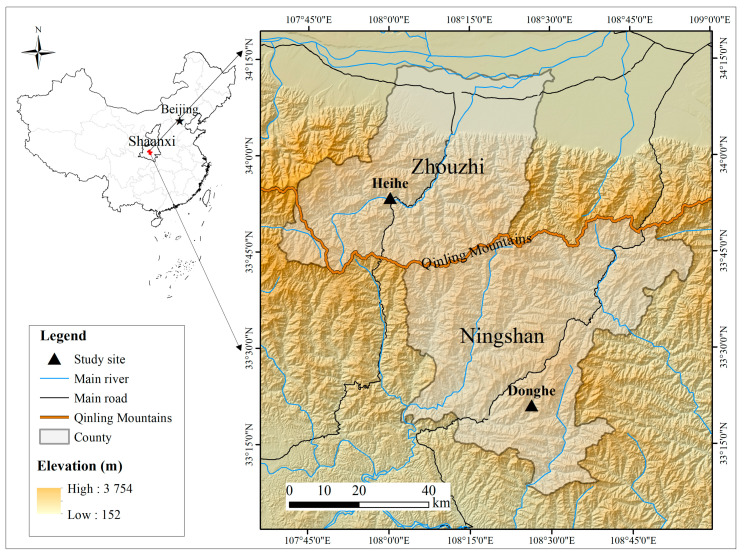
General location of Chinese Giant Salamanders reintroduction.

**Figure 2 animals-11-01480-f002:**
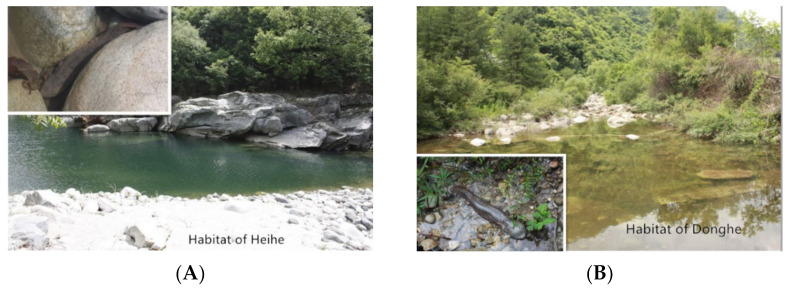
Environment of two sites where we reintroduced the Chinese giant salamander. (**A**) Habitat of the Heihe River; (**B**) Habitat of the Donghe River.

**Table 1 animals-11-01480-t001:** Mixed-effects logistic models ranked by Akaike’s information criterion (AIC).

River	Hypothesis	Model Structure	*K*	AIC	∆AIC
Donghe	Positive effect of moon phase	~moon + (1|ID)	3	4746.9	0
	Null	~(1|ID)	2	4751.8	4.9
	Positive effect of body condition	~condition + (1|ID)	3	4751.9	5
	Positive effect of precipitation	~precipitation + (1|ID)	3	4752.0	5.1
	Positive effect of precipitation in the previous day	~precipitation_pre_ + (1|ID)	3	4752.8	5.9
	Positive effect of body mass	~mass + (1|ID)	3	4753.6	6.7
	Positive effect of temperature	~temperature + (1|ID)	3	4753.7	6.8
Heihe	Null	~(1|ID)	2	1514.5	0
	Positive effect of temperature	~temperature + (1|ID)	3	1515.2	0.7
	Positive effect of moon phase	~moon + (1|ID)	3	1515.4	0.9
	Positive effect of precipitation in the previous day	~precipitation_pre_ + (1|ID)	3	1516.5	2
	Positive effect of precipitation	~precipitation + (1|ID)	3	1516.5	2
	Positive effect of body mass	~mass + (1|ID)	3	1516.5	2
	Positive effect of body condition	~condition + (1|ID)	3	1516.5	2

Explaining whether reintroduced giant salamanders moved at the Donghe River (*n* = 4252) and the Heihe River (*n* = 1239) based on body mass, body condition, temperature, moon phase, and precipitation, with salamander ID as a random effect.

**Table 2 animals-11-01480-t002:** Linear mixed-effects models ranked by Akaike’s information criterion (AIC).

River	Hypothesis	Model Structure	*K*	AIC	∆AIC
Donghe	Positive effect of temperature	~temperature + (1|ID)	4	22,098	0
	Positive effect of temperature and precipitation	~temperature + precipitation + (1|ID)	5	22,099	1
	Positive effect of precipitation	~precipitation + (1|ID)	4	22,127	29
	Null	~+ (1|ID)	3	22,129	31
	Positive effect of moon phase	~moon + (1|ID)	4	22,129	31
	Positive effect of precipitation in the previous day	~precipitation_pre_ + (1|ID)	4	22,129	31
	Positive effect of body mass	~mass + (1|ID)	4	22,131	33
	Positive effect of body condition	~condition + (1|ID)	4	22,131	33
Heihe	Positive effect of temperature	~temperature + (1|ID)	4	7921.5	0
	Positive effect of temperature and precipitation in the previous day	~temperature + precipitation_pre_ + (1|ID)	5	7921.5	0
	Positive effect of precipitation in the previous day	~precipitation_pre_ + (1|ID)	4	7928.6	7.1
	Null	~(1|ID)	3	7928.7	7.2
	Positive effect of body condition	~condition + (1|ID)	4	7928.8	7.3
	Positive effect of moon phase	~moon + (1|ID)	4	7929.2	7.7
	Positive effect of precipitation	~precipitation + (1|ID)	4	7929.7	8.2
	Positive effect of body mass	~mass + (1|ID)	4	7930.6	9.1

Explaining daily distance moved of reintroduced giant salamanders at the Donghe River (*n* = 3196) and the Heihe River (*n* = 870) based on body mass, body condition, temperature, moon phase, and precipitation, with salamander ID as a random effect.

## Data Availability

The data presented in this study are available on request from the corresponding author.

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
