# Peer review of "Abiotic and Biotic Influences on the Movement of Reintroduced Chinese Giant Salamanders (Andrias davidianus) in Two Montane Rivers"

_animals, 2021, doi:10.3390/ani11061480_

Round 1
Reviewer 1 Report
Dear Authors,
I very much enjoyed reading your manuscript - thank you. This is an interesting, well executed study that has been written up very clearly.
There are just a few points that need clarification. Please see below.
Line 17: remove 'its'
Line 18: remove 'So' and 'their'
Line 36: 'amixed' insert gap a_mixed
Line 94: '....res(e.g.,.......) insert gap '....res_(e.g.,......)
Line 109: 'overlonger' needs a gap over_longer
Materials and Methods, 2.1 Study area and animals - please could you comment on the rearing conditions - describe the type of enrichment and environmental resources present during rearing? Early life experience has profound influence on later behavioural development. Crucially were the rearing conditions the same or comparable between both older and younger animals?
Also, as well as being older or younger the animals are also wild hatched or captive hatched and while you can not obviously isolate the effects of this in your study it needs to be considered in the Discussion as a contributing factor.
What is prey density like at each release area? If you cannot comment on this directly, density of prey species and food availability therefore at both areas needs to be considered as a possible influential factor in the Discussion.
Line 220-221: what data presented suggests a negative effect of precipitation on the younger group? I may have misunderstood this but the patterns of effect in Table 2 suggest the same directionality of effect for all factors in both older and younger groups? Can you make this result clearer?
In the discussion generally please can you also discuss possible effects of other environmental differences between the two release sites - they are mentioned but how reasonable is it to consider these other factors may have or may not have a significant effect? What does literature on wild Chinese giant salamander suggest, or literature on similar species? Also please add in discussion on early life experience and any differences in rearing conditions (following on from comment made above).
Line 231: eso-predator, remove hyphen for consistency
Line 244: Barusbarbus needs a gap to read Barbus_barbus
Line 260: 'related' I think this should be 'relate' no d?
Reviewer 2 Report
General comment: The authors present a well written manuscript with new insight into the movement of giant salamanders. There is one inconsistency that needs to be dealt with. In the discussion (lines 220-221) authors say that precipitation had a positive effect on displacement of older salamanders, but a negative effect in the younger ones. However, in the conclusion (lines 284-285) authors say that precipitation had a positive effect on movement. Please clearly state what is being observed to prevent reader misinterpretation.
Line-by-line comments:
Page 2, line 46-47: Two keywords are also in the title, I suggest replacing them by other terms such as “demographics” and “tracking”.
Page 2, line 53: There is a space missing between the period and the word “For”.
Page 2, line 57: There is a space missing between the period and the word “Hence”.
Page 2, line 60: “animals” is plural, it should be “their internal star” not “its internal state”.
Page 2, line 61: There is a space missing between the period and the word “However”.
Page 2, line 65: Please replace “be affective on” by “to affect”. And remove “on” on “on others”.
Page 2, line 70: Please introduce the scientific species name here, also replace the first “and” by a comma.
Page 2, line 75: Please add a “being” in “species [being] listed”.
Page 2, line 74-75: Please add references to this sentence (for both the IUCN listing and the Chinese one).
Page 3, line 91: Please replace “Andrias” by “A.” as the genus name has been introduced (added on a previous comment).
Page 3, line 94: There is a space missing before the bracket.
Page 3, line 91 to 94: It is unclear what the authors mean by this sentence. Are the authors saying that studies on other species can inform on their target species? Please rephrase to make it clearer.
Page 3, line 97: Please add a “previously” to this sentence to clarify that these are not results from the study described in this manuscript.
Page 4, line 141: There is an extra period after “Figure 3”.
Page 5, line 151: There is an extra period after “Figure 4”
Page 5, line 159: Please cite ArcGIS.
Page 6, line 181: Please cite R.
Page 8, line 231: There is a space missing in the scientific name of the otter.
Page 8, line 244: There is a space missing in the scientific name of the barbel.
Page 8, line 257-258: The connector in the second clause does not make sense. Please replace “that river flow usually has” by “and it was found that usually river flow has”.
Page 8, line 258: There is a space missing between the period and “However”.
Page 8, line 160: There is a space missing before the bracket.
Page 8, line 260: Please replace “related” for “relate”.
Page 10, line 314: Please remove “ECOLOGY”.
References: Please reformat the references. There are multiple issues that need to be addressed such as species names not italicized, all words on title capitalized, periods missing after authors initials, double punctuation signs in a row.
Figure 1: Please add a thousand separator to the high elevation value. It is not clear what the small inset map is supposed to represent, it seems to me that it does not bring any information. Please remove. If authors meant to present something with it please explain what it is in the figure legend.
Table 1: Please add the thousand separator throughout the table and the table note.
Table 2: Please add the thousand separator throughout the table and table note.
Reviewer 3 Report
The manuscript „Abiotic and biotic influences on the movement of reintroduced Chinese giant salamanders (Andrias davidianus) in two montane rivers“ by Wang et al. is a neat study giving new insights into the movement patterns and their abiotic drivers of a highly endangered amphibian species. The introduction is well structured, and the goals are nicely explained. Also, the methods and the statistical results are presented well, and the discussion shows a good knowledge of the literature.
I am only missing one major detail: how much did the salamanders move? Even though the actual movement results were already presented in other publications, it would still be interesting to learn how big for example the effect of the moon was on the average moved distance.
Another issue I have are the figures: figure 1 would benefit from some more information (see below) and figures 3 and 4 seem rather unimportant. I would prefer a figure that actually shows some of the results, e.g. a graph showing the differences in moved average distance under different abiotic conditions.
Besides this, I only have some small comments:
- line 52: “is increasingly being used” – please delete being
- line 53, 57, 61: space behind dot is missing
- line 94: space before parenthesis is missing
- Figure 1: Just looking at the map, it does not become totally clear where the animals were reintroduced: generally in the shown area, or at the two red areas marked as study sites? The text clarifies this, but a better legend would be helpful. The same about the rivers: showing only main rivers, leaves it unclear (without further reading the text) that the site named Donghe is also a river/smaller stream. Furthermore, a kilometer-scale would make it easier to understand on first sight how big the shown area is
- line 127 following: a few sentences about the biology of the animals would be helpful. You are talking about 3 and 5 year old juveniles. At what age do they reach adulthood?
- figures 3 and 4 might be better placed into the supplementary materials. They do not seem specifically important for the manuscript
- line 187-188: the minimum temperature (or a temperature range) for both rivers may be interesting, too
- line 260: related should say relate
